# Optimal Planning as Constraint Optimization

## Enrico Giunchiglia, Armando Tacchella

DIBRIS - Università degli Studi di Genova
Viale Causa, 13 - 16145 Genova, Italy
enrico.giunchiglia@unige.it, armando.tacchella@unige.it

### Abstract

We consider the problem of optimal planning in deterministic domains and reduce it to the problem of finding an optimal solution of a corresponding constraint optimization problem incorporating a bound $n$ on the maximum length of the plan. By solving the latter, we can conclude whether $(i)$ the plan found is optimal even for bounds greater than $n$; or $(ii)$ we need to increase $n$; or $(iii)$ it is useless to increase $n$ since the planning problem has no solution. Our approach $(i)$ substantially generalizes previous approaches for optimal symbolic deterministic planning; $(ii)$ allows to compute non trivial lower bounds on the cost and length of optimal plans; and $(iii)$ produces an encoding linear in the size of the planning problem and the bound $n$.

## 1 Introduction

We consider the problem of optimal planning in deterministic domains. Given a planning problem $\Pi$ with costs $C$, We assume $(i)$ that $\Pi$ is specified with 3 formulas in conjunctive normal form (CNF) giving the initial state, valid transitions and goal states, and $(ii)$ that $C$ associates a non negative real number to every valid transition between two states. Our objective is to determine an optimal plan, i.e., a sequence of actions leading from the initial state to a goal state with minimum associated total cost, defined as the sum of the costs of the transitions induced by the actions in the plan.

In particular, we extend the planning as satisfiability approach (Kautz and Selman 1992) and reduce the problem of finding an optimal plan for $\langle \Pi, C \rangle$ to the one of solving a corresponding constraint optimization problem incorporating a bound $n$ on the maximum length of the plan. The basic idea is to construct an encoding $\Pi_n^O$ of $\Pi$ and $C_n^O$ of $C$ such that each valid plan $\pi$ of $\Pi$ with cost $C(\pi)$

- bijectively corresponds to a model $\pi_n^O$ of $\Pi_n^O$ having cost $C_n^O(\pi_n^O) = C(\pi)$, if $\pi$ has at most $n$ actions, and
- corresponds to a model $\pi_n^O$ of $\Pi_n^O$ having cost $C_n^O(\pi_n^O) \leq C(\pi)$, if $\pi$ has more than $n$ actions.

Thus, if $\pi_n^O$ is an optimal model of $\langle \Pi_n^O, C_n^O \rangle$ then

1. if $\pi_n^O$ corresponds to a plan $\pi$ of $\Pi$ with at most $n$ actions, then $\pi$ is an optimal plan of $\Pi$, and
2. if $\pi_n^O$ does not correspond to a plan of $\Pi$ with at most $n$ actions then we have to increase the bound $n$.

Moreover, if $\Pi_n^O$ is unsatisfiable then $\Pi$ does not admit a valid plan and it is useless to increase the bound $n$.

Since we place no restriction on the CNF formula specifying the valid transitions, our work substantially generalizes previous approaches for optimal symbolic deterministic planning. In particular, this paper builds on and significantly extends (Leofante et al. 2020) which is restricted to numeric planning problems expressible in PDDL2.1 level 2 (Fox and Long 2003). Despite being far more general than (Leofante et al. 2020), $(i)$ we provide non trivial lower bounds on the cost of the optimal plans and on the length of valid plans, and $(ii)$ our encoding never exponentially blows up since it is guaranteed to be linear in the size of $\Pi$ and the bound $n$.

The paper is structured as follows. After the formal framework, we focus on how to encode plans with length smaller than or equal to the bound (section 3), and then we consider plans longer than the bound (section 4). We put all the pieces together in section 5, ending the paper in section 6 with some final considerations, including related and future works.

## 2 Formal framework

We consider deterministic planning problems $(i)$ that can be described using finitely many state and action variables, and $(ii)$ whose initial state, valid transitions and goal states are the models of quantifier free CNF formulas. Thus, our approach is completely general and captures many logic based planning representation languages, like grounded PDDL 2.1 level 2 (Fox and Long 2003) and the action language $\mathcal{C}$ (Giunchiglia and Lifschitz 1998) in the deterministic case.

For the language signature, we assume to have

1. a non empty finite set $\mathcal{X}$ of *state variables*, each variable $x \in \mathcal{X}$ equipped with a *domain* $dom(x)$ representing the values the variable can assume,
2. a finite set $\mathcal{A}$ of Boolean *action variables*,
3. a copy $\mathcal{X}'$ of $\mathcal{X}$ of *next state variables* such that, for each state variable $x \in \mathcal{X}$, there is a corresponding variable $x' \in \mathcal{X}'$ with $dom(x') = dom(x)$.

An *assignment* to a set of variables $\mathcal{V}$ is a function mapping each variable in $\mathcal{V}$ to an element of its domain. In the case of Boolean variables, their domain is $\{\top, \bot\}$ for truth and falsity, and we use $v$ in place of $v = \top$. A *state* (resp. *action*, resp. *next state*) is an assignment to the variables $\mathcal{X}$ (resp. $\mathcal{A}$, resp. $\mathcal{X}'$). States, actions and next states are denoted with

$\sigma, \sigma_0, \ldots, \alpha, \alpha_0, \ldots,$ and $\sigma', \sigma'_0, \ldots$ respectively. A *transition* is an assignment to all the state, action and next state variables at hand. Besides variables, we assume to have other possibly theory dependent symbols (like "0", "+", "$\geq$") and auxiliary symbols (like "(" and ")") that are used to define atomic formulas, literals and well formed formulas. We take for granted standard logic notions like satisfiability, entailment, model, and the like. Unless explicitly specified, assignments are total. (Partial) actions are represented with the set of action literals they satisfy.

A *(deterministic) planning problem* is a 5 tuple $\Pi = \langle \mathcal{X}, \mathcal{A}, I(\mathcal{X}), T(\mathcal{X}, \mathcal{A}, \mathcal{X}'), G(\mathcal{X}) \rangle$ where

1. $I(\mathcal{X})$ is the *initial state formula* in the state variables $\mathcal{X}$, assumed to be satisfied by exactly one state;

2. $T(\mathcal{X}, \mathcal{A}, \mathcal{X}')$ is the *transition relation*, i.e., a formula in the $\mathcal{X}, \mathcal{A}, \mathcal{X}'$ variables, whose models are the *valid* transitions. For each state $\sigma$ and action $\alpha$ it is assumed that there is at most one valid transition $\sigma, \alpha, \sigma'$;

3. $G(\mathcal{X})$ is the *goal formula* in the state variables $\mathcal{X}$, whose models are the *goal states*.

Without loss of generality, we assume $I(\mathcal{X})$, $T(\mathcal{X}, \mathcal{A}, \mathcal{X}')$ and $G(\mathcal{X})$ to be in CNF, i.e, that each formula is a conjunction of clauses, where a *clause* is a disjunction of literals.

In the following, $lx, lx_1, \ldots$ (resp. $la, la_1, \ldots$, resp. $lx', lx'_1, \ldots$) denote state (resp. action, resp. next state) literals, i.e., literals in the $\mathcal{X}$ (resp. $\mathcal{A}$, resp. $\mathcal{X}'$) variables. When convenient, we use also the symbol "$\rightarrow$" for implication and write clauses in $T(\mathcal{X}, \mathcal{A}, \mathcal{X}')$ either as

$$\bigwedge_{i=1}^{p} la_i \rightarrow \bigvee_{i=1}^{q} lx_i \tag{1}$$

$(p, q \geq 0)$ to model that $(\bigvee_{i=1}^{q} lx_i)$ is an *explicit precondition* of the partial actions which satisfy $(\bigwedge_{i=1}^{p} la_i)$, or as

$$\bigwedge_{i=1}^{p} la_i \wedge \bigwedge_{i=1}^{q} lx_i \rightarrow \bigvee_{i=1}^{r} lx'_i \tag{2}$$

$(p, q \geq 0, r \geq 1)$, to model that $(\bigvee_{i=1}^{r} lx'_i)$ is an *explicit (conditional) effect* of the partial action $\{la_1, \ldots, la_p\}$ with the conditions in $\{lx_1, \ldots, lx_q\}$.

**Running Example** Consider a domain SQUARE in which a numeric variable $var$ is initialized to a fixed value $V_I \in \mathbb{R}$ and should reach a fixed value $V_G \in \mathbb{R}$. The value of $var$ can be changed only in states with $var \geq 0$, and in the next state the value of $var$ is automatically incremented by 1 unless it is squared. This domain can be formalized as the planning problem $\Pi = \langle \mathcal{X}, \mathcal{A}, I(\mathcal{X}), T(\mathcal{X}, \mathcal{A}, \mathcal{X}'), G(\mathcal{X}) \rangle$ where $\mathcal{X} = \{var\}$, $\mathcal{A} = \{square\}$, $I(\mathcal{X}) = (var = V_I)$, $G(\mathcal{X}) = (var = V_G)$, and $T(\mathcal{X}, \mathcal{A}, \mathcal{X}')$ is the formula

$$(\neg square \wedge var \geq 0 \rightarrow var' = var + 1) \wedge$$
$$(square \rightarrow var \geq 0) \wedge (square \rightarrow var' = var^2) \wedge \tag{3}$$
$$(var < 0 \rightarrow var' = var).$$

Indeed, SQUARE has been formalized as above to make the example simple yet illustrative for the theory below. $\square$

Let $\Pi = \langle \mathcal{X}, \mathcal{A}, I(\mathcal{X}), T(\mathcal{X}, \mathcal{A}, \mathcal{X}'), G(\mathcal{X}) \rangle$ be a planning problem. Our next step is to define the valid plans of

$\Pi$. We mostly use the terminology of (Fox and Long 2003; Haslum et al. 2019). If $F(\mathcal{V})$ is a formula/function in the $\mathcal{V}$ variables and $\mu$ is a partial assignment to $\mathcal{V}$ defined on $\mathcal{U} \subseteq \mathcal{V}$, by $F(\mu)$ we mean the formula/function obtained by substituting each variable $v \in \mathcal{U}$ with $\mu(v)$ in $F(\mathcal{V})$.

An action $\alpha$ is *executable* in a state $\sigma$ if there is a next state $\sigma'$ satisfying $T(\sigma, \alpha, \mathcal{X}')$, in which case the *result of executing $\alpha$ in $\sigma$* is the state $\sigma''$ such that, for each state variable $x$, $\sigma''(x) = \sigma'(x')$. A *plan* (*of length $k$*) is a sequence of $k \geq 0$ actions.

Consider a plan $\pi = \alpha_0; \ldots; \alpha_{k-1}$ $(k \geq 0)$. $\pi$ is *executable* if for each $i \in [0, k - 1]$, $\alpha_i$ is executable in $\sigma_i$, where

1. $\sigma_0$ is the state satisfying the initial state formula, and

2. $\sigma_{i+1}$ is the result of executing $\alpha_i$ in $\sigma_i$.

If $\pi$ is executable, the state $\sigma_i$ $(0 \leq i \leq k)$ as above defined is the *$i$-th state induced by $\pi$*. The plan $\pi$ is *valid* if it is executable and the $k$-th induced state $\sigma_k$ satisfies $G(\mathcal{X})$.

For the definition of optimal plan, we introduce a cost associated to each valid transition. By $C_{min}$ we denote a fixed positive constant. A pair $\langle \Pi, C \rangle$ is a *planning problem with costs* if $C$ is a *cost function* such that for each valid transition $\sigma, \alpha, \sigma'$, (i) $C(\sigma, \alpha, \sigma') \geq C_{min}$ whenever $\sigma'(x') \neq \sigma(x)$ for some state variable $x$, and (ii) $C(\sigma, \alpha, \sigma') \geq 0$ otherwise. If $\pi$ is a valid plan, the *cost $C(\pi)$* of $\pi$ is the sum of the costs of each transition, i.e.,

$$C(\pi) = \sum_{i=0}^{k-1} C(\sigma_i, \alpha_i, \sigma'_{i+1})$$

where $\sigma_i$ and $\sigma_{i+1}$ are the $i$-th and $(i + 1)$-th states induced by $\pi$ and, for each $x \in \mathcal{X}$, $\sigma'_{i+1}(x') = \sigma_{i+1}(x)$. The plan $\pi$ is *optimal* if it is valid and there is no valid plan with a smaller cost.

**Running Example** In SQUARE, we further assume that the cost of each transition is the maximum between 1 and the difference between the new and old values of $var$. Formally,

$$C(\mathcal{X}, \mathcal{A}, \mathcal{X}') = max(var' - var, 1).$$

Then, if $V_I = 1$ and $V_G = 9$, the plans $\xi = \{square\}; \{\neg square\}; \{\neg square\}; \{square\}$, and $\pi = \{\neg square\}; \{\neg square\}; \{square\}$, are both valid, but only $\pi$ is optimal (since $C(\xi) = 9$ and $C(\pi) = 8$), and there exist only two other optimal plans of length 7 and 8. $\square$

As a consequence of the assumption that every valid transition to a different state has an associated cost greater than or equal to $C_{min} > 0$, we have the following fact.

**Proposition 1** *Let $\langle \Pi, C \rangle$ be a planning problem with costs. If $\pi$ is a valid plan of $\Pi$ with cost $C(\pi)$ then there exists an optimal plan of length less than or equal to $\lfloor C(\pi)/C_{min} \rfloor$.*

## 3 Plans shorter than or equal to the bound

Let $\Pi = \langle \mathcal{X}, \mathcal{A}, I(\mathcal{X}), T(\mathcal{X}, \mathcal{A}, \mathcal{X}'), G(\mathcal{X}) \rangle$ be a planning problem with costs $C(\mathcal{X}, \mathcal{A}, \mathcal{X}')$, and let $n \geq 0$ be a fixed integer called *bound* or *number of steps*.

Following the planning as satisfiability approach (Kautz and Selman 1992), we make $n + 1$ disjoint copies

$\mathcal{X}_0, \ldots, \mathcal{X}_n$ of the set $\mathcal{X}$ of state variables, and $n$ copies $\mathcal{A}_0, \ldots, \mathcal{A}_{n-1}$ of the set $\mathcal{A}$ of action variables. Then, for each $i \in [0, n-1]$, $T(\mathcal{X}_i, \mathcal{A}_i, \mathcal{X}_{i+1})$ is the formula obtained substituting each variable $x \in \mathcal{X}$ (resp. $a \in \mathcal{A}$, $x' \in \mathcal{X}'$) with $x_i \in \mathcal{X}_i$ (resp. $a_i \in \mathcal{A}_i$, $x_{i+1} \in \mathcal{X}_{i+1}$) in $T(\mathcal{X}, \mathcal{A}, \mathcal{X}')$, and similarly for other formulas like $I(X_0)$, $G(\mathcal{X}_n)$ and $C(\mathcal{X}_i, \mathcal{A}_i, \mathcal{X}_{i+1})$.

Then, we define

$$\Pi_n^S = I(\mathcal{X}_0) \wedge \bigwedge_{i=0}^{n-1} T(\mathcal{X}_i, \mathcal{A}_i, \mathcal{X}_{i+1}) \wedge G(\mathcal{X}_n),$$
$$C_n^S = \sum_{i=0}^{n-1} C(\mathcal{X}_i, \mathcal{A}_i, \mathcal{X}_{i+1}).$$

Notice that both $\Pi_n^S$ and $C_n^S$ are in the variables $\mathcal{X}_0, \mathcal{A}_0, \ldots, \mathcal{X}_{n-1}, \mathcal{A}_{n-1}, \mathcal{X}_n$. $\Pi_n^S$ and $C_n^S$ define a constraint optimization problem, whose *optimal models* are the models of $\Pi_n^S$ that have minimum associated cost $C_n^S$.

**Lemma 1** *Let* $\Pi$ *be a planning problem. Let* $\pi = \alpha_0; \ldots; \alpha_{n-1}$ *be a plan of* $\Pi$. *There exists at most one model* $\pi_n^S$ *of* $\Pi_n^S$ *such that for each variable* $a_i \in \mathcal{A}_i$ $(0 \leq i < n)$, $\pi_n^S(a_i) = \alpha_i(a)$.

According to the lemma, for each plan $\pi$ we have at most one corresponding model $\pi_n^S$ of $\Pi_n^S$. Indeed, we have a tighter correspondence between the valid plans of $\Pi$ and the models of $\Pi_n^S$ and their respective costs.

**Proposition 2** *Let* $\langle \Pi, C \rangle$ *be a planning problem with costs. Let* $\pi$ *be a plan of length* $n$. $\pi$ *is a valid plan of* $\Pi$ *iff* $\pi_n^S$ *is a model of* $\Pi_n^S$, *and* $C(\pi) = C_n^S(\pi_n^S)$.

Notice that $\Pi_n^S$ and $C_n^S(\pi_n^S)$ encode the validity and the cost of plans of length exactly $n$. In order to consider also plans with length smaller than the bound, the transition relation $T(\mathcal{X}, \mathcal{A}, \mathcal{X}')$ may need to be modified in order to ensure $\Pi$ to be *inertial*, i.e., that for every state $\sigma$ there exists an action $\alpha$ whose execution in $\sigma$ results in the same state $\sigma$ with cost 0. To deal with inertia, we

1. extend the action signature with the variable *NoOp*, and
2. define $T^I(\mathcal{X}, \mathcal{A} \cup \{NoOp\}, \mathcal{X}')$ to be

$$T^I(\mathcal{X}, \mathcal{A} \cup \{NoOp\}, \mathcal{X}') = (\neg NoOp \rightarrow T(\mathcal{X}, \mathcal{A}, \mathcal{X}')) \wedge$$
$$\bigwedge_{x \in \mathcal{X}} (NoOp \rightarrow x' = x) \wedge$$
$$\bigwedge_{a \in \mathcal{A}} (NoOp \rightarrow \neg a).$$

Imposing in the definition above that all the action variables $a \in \mathcal{A}$ have to be false whenever *NoOp* is true allows to establish a one-to-one correspondence between the valid plans of $\Pi$ of length $k \leq n$ and the models of

$$\Pi_n^I = I(\mathcal{X}_0) \wedge \bigwedge_{i=0}^{n-1} T^I(\mathcal{X}_i, \mathcal{A}_i \cup \{NoOp_i\}, \mathcal{X}_{i+1})$$
$$\wedge \bigwedge_{i=0}^{n-2} (NoOp_i \rightarrow NoOp_{i+1}) \wedge G(\mathcal{X}_n).$$

The following lemma defines the assignment $\pi_n^I$ to the variables in $\Pi_n^I$ corresponding to a valid plan $\pi$ of length $k \leq n$.

**Lemma 2** *Let* $\Pi$ *be a planning problem. Let* $\pi = \alpha_0; \ldots; \alpha_{k-1}$ *be a plan of* $\Pi$ *of length* $k \leq n$. *There exists at most one model* $\pi_n^I$ *of* $\Pi_n^I$ *such that for each variable* $a_i \in \mathcal{A}_i$ $(0 \leq i < k)$, $\pi_n^I(a_i) = \alpha_i(a)$ *and* $\pi_n^I(NoOp_k) = \ldots = \pi_n^I(NoOp_{n-1}) = \bot$.

If we define $C^I(\mathcal{X}, \mathcal{A} \cup \{NoOp\}, \mathcal{X}')$ to be such that, for each assignment $\sigma, \alpha, \sigma'$ to $\mathcal{X}, \mathcal{A}, \mathcal{X}'$,

$$C^I(\sigma, \alpha \cup \{NoOp = \bot\}, \sigma') = C(\sigma, \alpha, \sigma'),$$
$$C^I(\sigma, \alpha \cup \{NoOp = \top\}, \sigma') = 0,$$

then we have also that the cost $C(\pi)$ of a plan $\pi$ of length $k \leq n$ is equal to $C_n^I(\pi_n^I)$, defined as:

$$C_n^I = \sum_{i=0}^{n-1} C^I(\mathcal{X}_i, \mathcal{A}_i \cup \{NoOp_i\}, \mathcal{X}_{i+1}).$$

**Proposition 3** *Let* $\langle \Pi, C \rangle$ *be a planning problem with costs. Let* $\pi$ *be a plan of length* $k \leq n$. $\pi$ *is a valid plan of* $\Pi$ *iff* $\pi_n^I$ *is a model of* $\Pi_n^I$, *and* $C(\pi) = C_n^I(\pi_n^I)$.

Owing to proposition 3, we know that if a model $\pi_n^I$ of $\Pi_n^I$ is optimal (i.e., all the other models $\rho_n^I$ of $\Pi_n^I$ are such that $C_n^I(\rho_n^I) \geq C_n^I(\pi_n^I)$), then there is no valid plan of $\Pi$ with length $\leq n$ and cost smaller than $C(\pi)$.

**Running Example** Assume that $V_I = 1$ and $V_G = 9$ in our $\langle \Pi, C \rangle$ formalization of the SQUARE domain. From the previous example, we know that there is an optimal plan $\pi$, plus two other, say $\omega$ and $\rho$, of length 3, 7 and 8, respectively. Assuming $n = 8$, from the proposition we can conclude that $\pi_n^I$, $\omega_n^I$ and $\rho_n^I$ are optimal models of $\langle \Pi_n^I, C_n^I \rangle$. On the other hand, from the fact that $\pi_n^I$, $\omega_n^I$ and $\rho_n^I$ are optimal models of $\langle \Pi_n^I, C_n^I \rangle$, the proposition does not allow us to conclude that $\pi$, $\omega$ and $\rho$ are optimal plans of $\langle \Pi, C \rangle$. $\square$

## 4  Plans longer than the bound

Let $\Pi = \langle \mathcal{X}, \mathcal{A}, I(\mathcal{X}), T(\mathcal{X}, \mathcal{A}, \mathcal{X}'), G(\mathcal{X}) \rangle$ be a planning problem with costs $C(\mathcal{X}, \mathcal{A}, \mathcal{X}')$, and let $n \geq 0$ be a bound. We build an abstract encoding $\Pi_n^A$ such that for each valid plan $\pi$ of length $k > n$ there is a corresponding model $\pi_n^A$ of $\Pi_n^A$ with cost $C_n^A(\pi_n^A) \leq C(\pi)$.

Consider a plan $\pi = \alpha_0; \ldots; \alpha_{k-1}$ of length $k > n$ and let $\sigma_n$ be the $n$-th state induced by $\pi$.

The definition of $\Pi_n^A$ is based on an abstract version $T^A(\mathcal{X}, \mathcal{BA}, \mathcal{BX}, \mathcal{V})$ of the transition relation $T(\mathcal{X}, \mathcal{A}, \mathcal{X}')$ and an abstract version $G^A(\mathcal{X}, \mathcal{BX}, \mathcal{V})$ of the goal condition $G(\mathcal{X})$, where

1. $\mathcal{BX}$ is a set containing one new Boolean *abstract state variable* $\overline{x}$ for each variable $x \in \mathcal{X}$: intuitively $\overline{x}$ is true in $\pi_n^A$ if $x$ is affected by some action $\alpha_i$ $(n \leq i < k)$;

2. $\mathcal{BA}$ is a set containing one new Boolean *abstract action variable* $\overline{la}$ for each action literal $la$ (thus, $|\mathcal{BA}| = 2 \times |\mathcal{A}|$): intuitively, $\overline{la}$ is true in $\pi_n^A$ if for some $n \leq i < k$, $\alpha_i(la) = \top$; and

3. $\mathcal{V}$ is a set of auxiliary Boolean variables necessary to maintain polynomial the size of $T^A(\mathcal{X}, \mathcal{BA}, \mathcal{BX}, \mathcal{V})$ and of its CNF conversion, and allowing the computation of non trivial lower bounds on the length and cost of $\pi$.

If $\pi$ is valid, then $\pi_n^A$ will be a model of both $T^A(\mathcal{X}, \mathcal{BA}, \mathcal{BX}, \mathcal{V})$ and $G^A(\mathcal{X}, \mathcal{BX}, \mathcal{V})$.

Similarly to $T(\mathcal{X}, \mathcal{A}, \mathcal{X}')$ (see eq. (1) and (2)), $T^A(\mathcal{X}, \mathcal{BA}, \mathcal{BX}, \mathcal{V})$ includes two types of clauses:

1. the abstract preconditions of each variable $\overline{la} \in \mathcal{BA}$, defined on the basis of a subset $\mathcal{P}^{la}$ of the preconditions of the partial action $\{la\}$ in $T(\mathcal{X}, \mathcal{A}, \mathcal{X}')$ and defining whether $\overline{la}$ can be set to true, and

2. the abstract effects affecting a variable $\overline{x} \in \mathcal{BX}$, defined on the basis of a superset $\mathcal{E}^x$ of the states and actions causing $x \in \mathcal{X}$ to change value (i.e., $x \neq x'$) and defining whether $\overline{x}$ has to be set to true given the $\mathcal{BA}$ variables set to true in the previous step.

Given that the abstract preconditions and effects involve the variables in the same set $\mathcal{BA}, \mathcal{BX}$, "loops" between the abstract preconditions and effects are possible. Such loops, if not ruled out, cause unwanted models, i.e., models not corresponding to plans executable starting from $\sigma_n$, the $n$-th state induced by $\pi$. In order to rule out such models, taking inspiration from (Janhunen 2004; Niemelä 2008), we impose a level ordering on $\mathcal{BA}, \mathcal{BX}$ ensuring that the first actions in $\mathcal{BA}$ set to true have their abstract preconditions satisfied by $\sigma_n$, and each variable in $\mathcal{BX}$ is not used to enable an abstract action in $\mathcal{BA}$ unless it has been previously set to true by some other abstract action in a lower level. This is obtained by introducing level ordering constraints associating a level in $[0, |\mathcal{X}|+1]$ to the following *level ordering variables* in $\mathcal{V}$:

1. $\lambda_{\overline{la}}$ for each abstract action $\overline{la} \in \mathcal{BA}$,

2. $\lambda_{\overline{x}}$ for each abstract state variable $\overline{x} \in \mathcal{BX}$,

3. $\lambda_p$ for each precondition $p \in \mathcal{P}^{la}$ of a partial action $\{la\}$,

4. $\lambda_e$ for each conjunction $e \in \mathcal{E}^x$ whose truth affects the value of $x$ in the resulting state,

5. $\lambda_{lx}$ for each state literal $lx$ in a conjunction $e \in \mathcal{E}^x$.

As part of $\mathcal{V}$, we also have one Boolean variable $\overline{e}$ for each conjunction $e \in \mathcal{E}^x$ and one additional Boolean variable $\overline{lx}$ for each state literal $lx$ in a conjunction $e \in \mathcal{E}^x$.

Consider an action literal $la$. The set of abstract preconditions of $\overline{la}$ are computed on the basis of a subset $\mathcal{P}^{la}$ of the preconditions of $la$. A disjunction $p$ of state literals is a *precondition* of an action literal $la$ if $T(\mathcal{X}, \mathcal{A}, \mathcal{X}')$ and the falsity of $p$ entails the falsity of $la$ (i.e., if $(T(\mathcal{X}, \mathcal{A}, \mathcal{X}') \wedge \neg p \wedge la)$ is unsatisfiable). Formally, the conjunction of the preconditions of $la$ is equivalent to

$$\exists \mathcal{A} \exists \mathcal{X}'(la \wedge T(\mathcal{X}, \mathcal{A}, \mathcal{X}')). \qquad (4)$$

**Running Example** The preconditions of *square* coincide with its explicit precondition ($var \geq 0$), while $\neg square$ has no preconditions, corresponding to the formula $\top$. Indeed, in this case, the explicit preconditions of *square* and $\neg square$ are equivalent to the formula (4). However, such equivalence in general does not hold since there can be also other implicit preconditions. For instance, if we add the clause ($var' > var$) to (3), the precondition of *square* becomes ($var > 1$), which indeed entails its explicit precondition. $\square$

Computing the preconditions of $la$ requires that the theory behind the planning problem admits a quantifier elimination procedure, but the existence of a quantifier free formula equivalent to (4) is in general undecidable since

$T(\mathcal{X}, \mathcal{A}, \mathcal{X}')$ can be an arbitrary Diophantine equation in $\mathcal{X}'$ (see (Helmert 2002)). However, there are cases in which such quantifier elimination is possible, though computationally expensive, e.g., using Fourier–Motzkin procedure, assuming variables are either Boolean or range over the reals, and that in $T(\mathcal{X}, \mathcal{A}, \mathcal{X}')$ there are only Boolean variables and linear inequalities. Furthermore, in many cases all preconditions are explicit in $T(\mathcal{X}, \mathcal{A}, \mathcal{X}')$, e.g., for PDDL encoded problems. Finally, in all cases — since we wish to compute a superset of the set of actions literals $la$ which have their precondition satisfied — we do not need all the preconditions of $la$, and we can just consider the explicit ones in $T(\mathcal{X}, \mathcal{A}, \mathcal{X}')$.

Given the last point, consider a subset $\mathcal{P}^{la}$ of the preconditions of $la$, which contains at least the explicit preconditions of $\{la\}$ in $T(\mathcal{X}, \mathcal{A}, \mathcal{X}')$ and thus also the clauses in $T(\mathcal{X}, \mathcal{A}, \mathcal{X}')$ without action and next state variables. Formally, let $\mathcal{P}^{la}$ be a set of disjunctions of state literals, each entailed by (4). Then, for each precondition $p = (\bigvee_{i=1}^{q} lx_i) \in \mathcal{P}^{la}$ ($q \geq 0$) in the state variables $\{x_1, \ldots, x_m\} \subseteq \mathcal{X}, T^A(\mathcal{X}, \mathcal{BA}, \mathcal{BX}, \mathcal{V})$ includes the clause

$$\overline{la} \rightarrow p \vee \bigvee_{i=1}^{m} \overline{x_i}.$$

The above clause models the fact that we consider the abstract precondition corresponding to $p$ satisfied, if $p$ is either satisfied by $\sigma_n$ or if one of its state variables has been affected by an abstract action at a lower level. For the level ordering constraint, we impose that the level $\lambda_p$ associated to $p$ is 0 if $p$ is satisfied by $\sigma_n$, and is the minimum of the levels associated to $\overline{x_1}, \ldots, \overline{x_m}$ and $|\mathcal{X}| + 1$ otherwise:

$$\bigvee_{i=1}^{q} lx_i \rightarrow \lambda_p = 0,$$
$$\bigwedge_{i=1}^{q} \neg lx_i \rightarrow \lambda_p = min(\lambda_{\overline{x_1}}, \ldots, \lambda_{\overline{x_m}}, |\mathcal{X}| + 1).$$

Then, the level $\lambda_{\overline{la}}$ associated to $\overline{la} \in \mathcal{BA}$ is the maximum of the levels associated to all the preconditions in $\mathcal{P}^{la}$ and 0, and $\overline{la}$ can be set to true only if its level is not $|\mathcal{X}| + 1$:

$$\lambda_{\overline{la}} = max(\lambda_p : p \in \mathcal{P}^{la}, 0), \quad \overline{la} \rightarrow \lambda_{\overline{la}} \neq |\mathcal{X}| + 1. \quad (5)$$

Now we consider the problem of computing the abstract effects, determining when an abstract state variable $\overline{x} \in \mathcal{BX}$ can be set to true. Consider a state variable $x$. Our goal is to set $\overline{x}$ to true when there is a state and an action which cause $x$ to change value in the resulting state. Such states and actions are those that satisfy

$$\exists \mathcal{X}'(x' \neq x \wedge T(\mathcal{X}, \mathcal{A}, \mathcal{X}')). \qquad (6)$$

As for the preconditions of an action literal, computing a quantifier free formula equivalent to the above may not be possible. However, we need to find a superset of the set of next state variables $x'$ which change value, and we can consider a superset of the desired states and actions. Thus, we can take $\mathcal{E}^x$ to be the set of the antecedents of the explicit effects (2) in $T(\mathcal{X}, \mathcal{A}, \mathcal{X}')$ such that

1. $x'$ occurs in a next state literal $lx_i'$ ($1 \leq i \leq r$), and

2. $(\bigwedge_{j=1}^{q} lx_j \wedge lx_i')$ does not entail $x' = x$.[1]

Consider a set $\mathcal{E}^x$ of conjunctions of state and action literals such that if $x$ changes value in the resulting state then at least one of the conjunctions in $\mathcal{E}^x$ is satisfied. Let $\mathcal{E}^x$ be a set of conjunctions of state and action literals such that the disjunction of the conjunctions in $\mathcal{E}^x$ is entailed by (6). Then, $T^A(\mathcal{X}, \mathcal{BA}, \mathcal{BX}, \mathcal{V})$ includes the following clauses:

1. for each conjunction $e = \bigwedge_{i=1}^{p} la_i \wedge \bigwedge_{i=1}^{q} lx_i \in \mathcal{E}^x$ $(p, q \geq 0)$, the clauses corresponding $(i)$ to
$$\overline{lx_i} \leftrightarrow lx_i \vee \bigvee_{j=1}^{m} \overline{x_j},$$
for each state literal $lx_i$ $(1 \leq i \leq q)$ in $e$ in the state variables $\{x_1, \ldots, x_m\} \subseteq \mathcal{X}$, and $(ii)$ to
$$\overline{e} \leftrightarrow \bigwedge_{i=1}^{p} \overline{la_i} \wedge \bigwedge_{i=1}^{q} \overline{lx_i},$$
all clauses modeling the fact that we consider $\overline{e}$ to be satisfied when the abstract version of the actions and conditions in $e$ are satisfied; and

2. the clauses saying that $\overline{x}$ is true iff one of abstract formulas in $\mathcal{E}^x$ is satisfied, equivalent to:
$$\overline{x} \leftrightarrow \bigvee_{e \in \mathcal{E}^x} \overline{e}.$$

For the level ordering constraint, we impose that

1. for each conjunction $e = \bigwedge_{i=1}^{p} la_i \wedge \bigwedge_{i=1}^{q} lx_i \in \mathcal{E}^x$ $(p, q \geq 0)$, $(i)$ that the level $\lambda_{lx_i}$ $(1 \leq i \leq q)$ associated to the state literal $lx_i$ in the state variables $x_1, \ldots, x_m$ $(m \geq 0)$ is 0 if $lx_i$ is satisfied in $\sigma_n$, and is the minimum of the levels associated to the abstract state variables $\{\overline{x_1}, \ldots, \overline{x_m}\}$ and $|\mathcal{X}| + 1$ otherwise:
$$lx_i \rightarrow \lambda_{lx_i} = 0,$$
$$\neg lx_i \rightarrow \lambda_{lx_i} = min(\lambda_{\overline{x_1}}, \ldots, \lambda_{\overline{x_m}}, |\mathcal{X}| + 1),$$
and $(ii)$ that the level $\lambda_e$ of $e$ is the maximum of the levels of the conditions and action literals in $e$ and 0:
$$\lambda_e = max(\lambda_{la_1}, \ldots, \lambda_{la_p}, \lambda_{lx_1}, \ldots, \lambda_{lx_q}, 0),$$

2. that the level $\lambda_{\overline{x}}$ of $\overline{x} \in \mathcal{BX}$ is 1 plus the minimum of the levels associated to each effect $e \in \mathcal{E}^x$ and $|\mathcal{X}|$, and $\overline{x}$ can be set to true only if its level is not $|\mathcal{X}| + 1$:
$$\lambda_{\overline{x}} = min(\lambda_e : e \in \mathcal{E}^x, |\mathcal{X}|) + 1,$$
$$\overline{x} \rightarrow \lambda_{\overline{x}} \neq |\mathcal{X}| + 1. \tag{7}$$

$T^A(\mathcal{X}, \mathcal{BA}, \mathcal{BX}, \mathcal{V})$ is the conjunction of the clauses associated to $\mathcal{P}^{la}$ and $\mathcal{E}^x$, for each action literal $la$ and state variable $x$.

**Running Example** Let $\mathcal{P}^{square} = \{var \geq 0\}$ and $\mathcal{P}^{\neg square} = \emptyset$, corresponding to the explicit preconditions of $\{square\}$ and $\{\neg square\}$ respectively. Let $\mathcal{E}^{var} = \{(\neg square \wedge var \geq 0), square\}$, corresponding to the first and third clauses in (3). Then, from $T^A(\mathcal{X}, \mathcal{BA}, \mathcal{BX}, \mathcal{V})$, it follows that $(|\mathcal{X}| = 1)$

---

[1] Such condition can be easily checked when $(i)$ $lx'$ is $x' = x$ or $x = x'$ or there is a conjunct $lx_i$ $(0 < i \leq q)$ equal to $x = v$ and $lx'$ is $x' = v$ (as it is the case in the explanatory and classical frame axioms of classical Boolean planning problems).

1. if $(var \geq 0)$ is false then, given (5) and (7), $\overline{\neg square}$ can be set to true but $\overline{square}$ and $\overline{var}$ are necessarily false since $\lambda_{\overline{square}} = \lambda_{\overline{var}} = 2, \lambda_{\overline{\neg square}} = 0$,

2. if $(var \geq 0)$ is true then $\overline{square}, \overline{\neg square}$ and $\overline{var}$ can be set to true since $\lambda_{\overline{square}} = \lambda_{\overline{\neg square}} = 0, \lambda_{\overline{var}} = 1$.

$\square$

Now we consider the definition of $G^A(\mathcal{X}, \mathcal{BX}, \mathcal{V})$, the abstract version of the goal formula $G(\mathcal{X})$. Consider the goal formula $G(\mathcal{X}) = \bigwedge_{i=1}^{s} \bigvee_{j=1}^{s_i} lx_{ij}$. $G^A(\mathcal{X}, \mathcal{BX}, \mathcal{V})$ is the CNF formula consisting of

1. for each clause $c_i = \bigvee_{j=1}^{s_i} lx_{ij}$ in the state variables $x_1, \ldots, x_m$ $(m \geq 0)$, the clauses corresponding to
$$c_i \vee \bigvee_{j=1}^{m} \overline{x_j}, \qquad \bigvee_{j=1}^{s_i} lx_{ij} \rightarrow \lambda_{c_i} = 0,$$
$$\bigwedge_{j=1}^{s_i} \neg lx_{ij} \rightarrow \lambda_{c_i} = min(\lambda_{\overline{x_1}}, \ldots, \lambda_{\overline{x_m}}, |\mathcal{X}| + 1),$$
where $\lambda_{c_i}$ is a new level ordering variable in $\mathcal{V}$, and

2. the clause ($\lambda_G$ is the last new variable in $\mathcal{V}$ we introduce)
$$\lambda_G = max(\lambda_{c_1}, \ldots, \lambda_{c_s}, 0).$$

The definition of the level ordering $\lambda_G$ associated to the goal formula allows us to define $(i)$ a lower bound $\lambda_G$ on the number of steps necessary, starting from the $n$-th induced state $\sigma_n$, to reach a goal state, and $(ii)$ a lower bound
$$C_n^G = \lambda_G \times C_{min}$$
of the cost to reach a goal state starting from $\sigma_n$.

We can state the desired correspondence between the plan $\pi$ with cost $C(\pi)$ and a model $\pi_n^A$ of $\Pi_n^A$ with cost $C_n^A(\pi_n^A)$. $\Pi_n^A$ and $C_n^A$ are defined below, while $\pi_n^A$ will be characterized with a lemma as we did for $\pi_n^I$ in Section 3.

$$\Pi_n^A = I(\mathcal{X}_0) \wedge \bigwedge_{i=0}^{n-1} T(\mathcal{X}_i, \mathcal{A}_i, \mathcal{X}_{i+1})$$
$$\wedge T^A(\mathcal{X}_n, \mathcal{BA}, \mathcal{BX}, \mathcal{V}) \wedge G^A(\mathcal{X}_n, \mathcal{BX}, \mathcal{V}),$$
$$C_n^A = C_n^S + C_n^G.$$

**Lemma 3** *Let $\Pi$ be a planning problem. Let $\pi = \alpha_0; \ldots; \alpha_{k-1}$ be a plan of $\Pi$ of length $k > n$. There exists at most one model $\pi_n^A$ of $\Pi_n^A$ such that*

1. *for each variable $a_i \in \mathcal{A}_i$ $(0 \leq i < n)$, $\pi_n^A(a_i) = \alpha_i(a)$,*
2. *for each action literal $la$, $\pi_n^A(\overline{la}) = \top$ iff there exists an action $\alpha_i$ with $i \in [n, k-1]$ and $\alpha_i(la) = \top$.*

**Proposition 4** *Let $\langle \Pi, C \rangle$ be a planning problem with costs. Let $G$ be the goal formula in $\Pi$. Let $\pi$ be a valid plan of length $k > n \geq 0$. Then, $\pi_n^A$ is a model of $\Pi_n^A$, $C(\pi) \geq C_n^A(\pi_n^A)$, and $k \geq n + \pi_n^A(\lambda_G)$.*

**Running Example** $G^A(\mathcal{X}, \mathcal{BX}, \mathcal{V})$ simplifies to
$$(var = V_G \vee \overline{var}) \wedge (var = V_G \rightarrow \lambda_G = 0) \wedge$$
$$(var \neq V_G \rightarrow \lambda_G = \lambda_{\overline{var}}).$$

Assuming that $V_G > V_I \geq 0$, then, for $n = 0$, for any model $\pi_n^A$ of $\Pi_n^A$, $\pi_n^A(\lambda_G) = 1 = |\mathcal{X}|$, meaning that, for $n = 0$, we can conclude that the length of each valid plan has 1 as lower bound. This is because, for every $n$, if $\Pi_n^A$ is satisfiable then it is always the case that $\lambda_G \leq |\mathcal{X}|$, and we have $|\mathcal{X}| = 1$. If we consider the planning problem with

$m$ Boolean state variables $\mathcal{X} = \{v_1, \ldots, v_m\}$ and no action variables, assuming that $I(\mathcal{X}) = \bigwedge_{i=1}^{m} \neg v_i$, $G(\mathcal{X}) = v_m$ and that the transition relation is a CNF formula equivalent to

$$v_1' \wedge \bigwedge_{i=1}^{m-1}(v_i \to v_{i+1}') \wedge \bigwedge_{i=1}^{m-1}(\neg v_i \to v_{i+1}' \leftrightarrow v_{i+1})$$

then valid plans (consisting of sequences of empty actions) have length $\geq m$ and, for $n = 0$, $\lambda_G = |\mathcal{X}|$. $\quad \square$

## 5 Optimal planning as Constraint Optimization

Let $\Pi = \langle \mathcal{X}, \mathcal{A}, I(\mathcal{X}), T(\mathcal{X}, \mathcal{A}, \mathcal{X}'), G(\mathcal{X}) \rangle$ be a planning problem with costs $C(\mathcal{X}, \mathcal{A}, \mathcal{X}')$, and let $n \geq 0$ be a bound. We combine the results in Sections 3, 4 and define a constraint optimization problem $\langle \Pi_n^O, C_n^O \rangle$ allowing to determine $(i)$ an optimal plan of length $k \leq n$, or $(ii)$ the non existence of a valid plan, or $(iii)$ whether the bound $n$ needs to be increased. These statements are consequences of the Theorem below, based on the following definition of $\langle \Pi_n^O, C_n^O \rangle$:

$$\begin{aligned}
\Pi_n^O = \ & I(\mathcal{X}_0) \wedge \bigwedge_{i=0}^{n-1} T^I(\mathcal{X}_i, A_i \cup \{NoOp_i\}, \mathcal{X}_{i+1}) \\
& \wedge \bigwedge_{i=0}^{n-2}(NoOp_i \to NoOp_{i+1}) \\
& \wedge T^A(\mathcal{X}_n, \mathcal{BA}, \mathcal{BX}, \mathcal{V}) \wedge G^A(\mathcal{X}_n, \mathcal{BX}, \mathcal{V}) \\
& \wedge (NoOp_{n-1} \to \lambda_G = 0) \\
& \wedge \bigwedge_{\overline{la} \in \mathcal{BA}}(\lambda_G = 0 \to \neg \overline{la}), \\
C_n^O = \ & C_n^I + C_n^G.
\end{aligned}$$

**Lemma 4** *Let $\Pi$ be a planning problem. Let $\pi = \alpha_0; \ldots; \alpha_{k-1}$ be a plan of $\Pi$. There exists at most one model $\pi_n^O$ of $\Pi_n^O$ such that, if $m = min(k, n)$,*
1. *for each variable $a_i \in \mathcal{A}_i$ $(0 \leq i < m)$, $\pi_n^O(a_i) = \alpha_i(a)$ and $\pi_n^O(NoOp_m) = \ldots = \pi_n^O(NoOp_{n-1}) = \bot$,*
2. *for each action literal $la$, $\pi_n^O(\overline{la}) = \top$ iff there exists an action $\alpha_i$ with $i \in [m, k-1]$ and $\alpha_i(la) = \top$.*

**Theorem 1** *Let $\langle \Pi, C \rangle$ be a planning problem with costs.*
1. *A plan $\pi$ of length $k$ is optimal iff there exists a bound $n \geq k$ such such that $\pi_n^O$ is an optimal model of $\langle \Pi_n^O, C_n^O \rangle$ and $\pi_n^O(\lambda_G) = 0$.*
2. *For a bound $n \geq 0$, if $\pi_n^O$ is an optimal model of $\langle \Pi_n^O, C_n^O \rangle$ and $\pi_n^O(\lambda_G) = 0$, then for every $m \geq n$, $\pi_m^O$ is an optimal model of $\langle \Pi_m^O, C_m^O \rangle$ and $\pi_m^O(\lambda_G) = 0$.*
3. *For a bound $n \geq 0$, if $\Pi_n^O$ is unsatisfiable then for every $m \geq n$, $\Pi_m^O$ is unsatisfiable and $\Pi$ has no valid plans.*
4. *For a bound $n \geq 0$, if $\pi_n^O$ is an optimal model of $\langle \Pi_n^O, C_n^O \rangle$ then any valid plan of $\Pi$ has cost greater than or equal to $C_n^O(\pi_n^O)$.*
5. *For a bound $n \geq 0$, if $\pi_n^O$ is an optimal model of $\langle \Pi_n^O, C_n^G \rangle$ and $\pi_n^O(\lambda_G) \neq 0$ then any valid plan of $\Pi$ has length greater than or equal to $(n + \pi_n^O(\lambda_G))$.*
6. *The size of $\Pi_n^O$ is $\mathcal{O}([\Pi] \times n)$, where $[\Pi]$ is the size of $\Pi$.*

Given Proposition 1, the Theorem guarantees that, assuming the existence of a valid plan for $\Pi$, we are able to determine an optimal plan by repeatedly solving the constraint optimization problem $\langle \Pi_n^O, C_n^O \rangle$ for increasing $n$, till an optimal model $\pi_n^O$ is found with $\pi_n^O(\lambda_G) = 0$. The second and third statements imply that we do not need to increment the bound in unitary steps: indeed, we can fix the new

bound according to some policy (see, e.g., (Rintanen, Heljanko, and Niemelä 2006; Rintanen 2012)). The fourth and fifth statements provide the lower bounds on the cost and length of valid plans. Notice that if $\pi_n^O$ is an optimal model of $\langle \Pi_n^O, C_n^O \rangle$ and $\pi_n^O(\lambda_G) \neq 0$, we can conclude neither the existence of a valid plan nor that valid plans have length $\geq n + \pi_n^O(\lambda_G)$. Indeed, the latter holds (fifth statement) assuming that the cost function of the optimization problem is fixed to $C_n^G$ (and not to $C_n^O = (C_n^I + C_n^G)$). Finally, the last statement ensures that our encoding is linear in the size of $\Pi$ and $n$.

**Running Example** If $V_I < 0$ and $V_G \neq V_I$ then, for any $n \geq 0$, $\Pi_n^O$ is unsatisfiable and indeed $\Pi$ does not have valid plans. If $V_I = 1$ and $V_G = 9$ there are three optimal plans of length 3, 7 and 8; and $(i)$ for $n \leq 6$, $\Pi_n^O$ has one optimal model with cost $(n-1)$ and satisfying $\lambda_G = 1$; $(ii)$ for $n = 7$, $\Pi_n^O$ has 3 optimal models with cost 8 but only two of them satisfy $\lambda_G = 0$; and $(iii)$ for $n \geq 8$, there are 3 optimal models and all of them satisfy $\lambda_G = 0$. If we extend the transition relation (3) with the constraint $(var < 9)$ and $V_I = 1$ and $V_G = 10$, then $\Pi_n^O$ admits one optimal model satisfying $\lambda_G = 1$ for $n \leq 8$, while for $n \geq 9$, $\Pi_n^O$ is unsatisfiable, proving that $\Pi$ has no valid plan. $\quad \square$

As the above example makes clear, it is possible to have $(i)$ a bound $n$ greater than the length of an optimal plan $\pi$ and $\pi_n^O$ is not an optimal model of $\langle \Pi_n^O, C_n^O \rangle$; $(ii)$ a bound $n$ for which we have various optimal models of $\langle \Pi_n^O, C_n^O \rangle$ but only some of them correspond to optimal plans; and $(iii)$ a bound $n$ after which for every optimal plan $\pi$, $\pi_n^O$ is an optimal model of $\langle \Pi_n^O, C_n^O \rangle$. It is also possible that the optimization problem $\langle \Pi_n^O, C_n^O \rangle$ becomes unsatisfiable for bounds greater than a certain value.

## 6 Conclusions, related and future work

We have shown how to reduce an optimal planning problem in deterministic domains with finitely many variables to a constraint optimization one. We have considered the problem in its full generality, making no other assumption about the domain. Our results are thus applicable to planning problems specified, e.g., in various versions of the PDDL language (in particular, in subsets of PDDL 2.1, 2.2, 3.1) and in the action language $\mathcal{C}$ when the domain is deterministic. We are not aware of comparable approaches as general as ours. Previous attempts to find solutions for optimal planning problems include (Robinson et al. 2010), where partial weighted MaxSAT is proposed as a backed to solve specific kinds of optimal planning problems. More recently, in (Davies et al. 2016) a mixed-integer programming encoding of a perfect heuristic is developed, landing on an incremental Boolean satisfiability encoding, while our results can be applied to back-ends dealing with decidable first order theories, e.g., satisfiability modulo theories. As for lower bounds, some results related to ours can be found in (Haslum 2012) presenting incremental lower bounds, but limited to additive cost planning problems, and (Haslum 2013) discussing optimal planning with conditional effects using a mechanism of relaxation similar to ours. Finally, some work closely related to ours can be found also in (Abdulaziz 2021) where upper bounds on the length of cost optimal plans that

are valid for problems with 0-cost actions are investigated. More in general, there are many papers focusing on optimal planning and/or showing how to translate planning problems in logic-based formalisms (see, e.g., (Ghallab, Nau, and Traverso 2004) for an overview). As mentioned, our work generalizes (Leofante et al. 2020) which considers numeric problems specified in PDDL 2.1 level 2. If we do not take into account the optimizations introduced by (Leofante et al. 2020) that are possible because of the restricted language used, the substantial difference is in the encoding of plans longer than the bound. In particular, to eliminate the unwanted models caused by loops between preconditions and effects, we use level order formulas based on (Janhunen 2004; Niemelä 2008), while Leofante et al. use loop formulas based on (Lin and Zhao 2002). However, with loop formulas $(i)$ the size of the encoding may exponentially blow up (Lifschitz and Razborov 2006), and $(ii)$ it is not possible to compute non trivial lower bounds of the length of valid plans and of their cost.

The primary extension of this work is to assess whether the proposed theory and/or a generalization/specialization scales in practice, also compared to other approaches. The results in (Leofante et al. 2020), but also in (Piacentini et al. 2018) for numeric problems, are encouraging even for sequential planning problems in which, in every action, at most one variable is true. Indeed, in the non sequential case, planners based on search have to evaluate $2^{|A|}$ possible actions in every state, making symbolic approaches like ours very appealing.

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
