# OpenReview forum: "Optimal Planning as Constraint Optimization"
_icaps-conference.org/ICAPS/2022/Workshop/HSDIP — HSDIP 2022_

### Official Review · Reviewer_S8q9 · 2022-04-21
**Missing technical details, missing empirical evaluation, nevertheless, interesting and relevant**

**Confidence:** 4
**Overall Score:** Accept

**Review:**

This paper considers the problem of optimal planning in deterministic domains. Given a planning problem Π with costs C, the objective is to determine an optimal plan, from the initial state to a goal state with minimum associated total cost. The approach for solving this problem is by a reduction to the problem of finding an optimal solution of a corresponding constraint optimization problem (bound n on the maximum length of the plan). In the rest of the paper, they show proper encoding and illustrate it with running examples.

The topic of the paper is relevant to the workshop, clear planning paper, an interesting approach, and well-established related work.

The paper is mostly well written and provides a good overall formal framework to introduce the problem, however, I found the frequent use of itemized points (i)(iI)(iii) in the paper a bit exhausting and interfering with the flow of the paper. This is a meter of style but in my opinion, sometimes it turns out as a list and not as a cohesive story.

The structure of the paper makes a lot of sense, starting with formally introducing the problem, moving to the encoding, and finally assembling all the pieces together. In particular, the running examples are very useful, adding intuition to the reading. However, I found the lack of proofs concerning. There is a substantial amount of technical results in this paper with no proper proof or proof guidelines. Some Lemmas /Propositions could be easily defined as Observations instead, and then be supported by a textual explanation, but as long as the authors decide to use Lemmas/ Propositions/ Theorems it is required to add a convincing proof even as supplementary material.

Another issue that I found concerning is that no empirical evaluations are presented. In the paper by Leofante et al. 2020, they show different domains in which the technique can be examined, in this paper however no experiment was reported.

---

### Official Review · Reviewer_bQPt · 2022-04-25
**Interesting preliminary work on optimal planning based on COP**

**Confidence:** 4
**Overall Score:** Accept

**Review:**

The paper is focused on optimal deterministic planning by reformulation to COP with a finite horizon bound (along the lines of "planning as satisfiability"). The proposed method seems to be very general in a sense that it can be used to classical planning problems as well as numerical planning and, interestingly, it can even prove unsolvability for bounds set high enough.

The paper is definitely not an easy read, but it is interesting and shows very promising idea. I could not really verify the technical soundness because it completely lacks proofs, but everything seems to be correct (at least intuitively it makes sense to me). It misses experimental evaluation, but authors seem to be determined to continue working on it. So, I think the paper should be accepted because it can result in interesting discussions during the workshop further encouraging authors to work in this direction.

I have one question which is maybe too far fetched at this point: Given the generality of the method, do you think there is a possibility of extending this work also to lifted planning/grounding? That is, instead of having just one bound on the length of the plan, to add another bound on the "breath" of the representation corresponding to the number ground atoms/actions and then ask COP solver to find both the plan and the (minimal?) grounding necessary to prove the optimality of the plan?

MInor issues:

page 2, third paragraph, right column: Shouldn't that be "k > 0" instead of "k >= 0"? Or do you allow empty plans?

page 3: "proposition (3)" -> "Proposition 3"